# Household Microenvironment and Under-Fives Health Outcomes in Uganda: Focusing on Multidimensional Energy Poverty and Women Empowerment Indices

**DOI:** 10.3390/ijerph19116684

**Published:** 2022-05-30

**Authors:** Zelalem G. Terfa, Sayem Ahmed, Jahangir Khan, Louis W. Niessen

**Affiliations:** 1Department of Clinical Sciences, Liverpool School of Tropical Medicine, Pembroke Place, Liverpool L3 5QA, UK; 2Centre for Environment and Development, College of Development Studies, Addis Ababa University, Addis Ababa P.O. Box 1176, Ethiopia; 3Department of International Public Health, Liverpool School of Tropical Medicine, Liverpool L3 5QA, UK; sayem.ahmed@lstmed.ac.uk (S.A.); jahangir.khan@gu.se (J.K.); louis.niessen@lstmed.ac.uk (L.W.N.); 4School of Public Health and Community Medicine, University of Gothenburg, 40530 Gothenburg, Sweden; 5Department of International Health, Johns Hopkins School of Public Health, Baltimore, MD 21205, USA

**Keywords:** multidimensional energy poverty, women empowerment, water, sanitation, acute respiratory infection, stunting, diarrhoea

## Abstract

Young children in low- and middle-income countries (LMICs) are vulnerable to adverse effects of household microenvironments. The UN Sustainable Development Goals (SDGs)—specifically SDG 3 through 7—urge for a comprehensive multi-sector approach to achieve the 2030 goals. This study addresses gaps in understanding the health effects of household microenvironments in resource-poor settings. It studies associations of household microenvironment variables with episodes of acute respiratory infection (ARI) and diarrhoea as well as with stunting among under-fives using logistic regression. Comprehensive data from a nationally representative, cross-sectional demographic and health survey (DHS) in Uganda were analysed. We constructed and applied the multidimensional energy poverty index (MEPI) and the three-dimensional women empowerment index in multi-variate regressions. The multidimensional energy poverty was associated with higher risk of ARI (OR = 1.32, 95% CI 1.10 to 1.58). Social independence of women was associated with lower risk of ARI (OR= 0.91, 95% CI 0.84 to 0.98), diarrhoea (OR = 0.93, 95% CI 0.88 to 0.99), and stunting (OR = 0.83, 95% CI 0.75 to 0.92). Women’s attitude against domestic violence was also significantly associated with episodes of ARI (OR = 0.88, 95% CI 0.82 to 0.93) and diarrhoea (OR = 0.89, 95% CI 0.84 to 0.93) in children. Access to sanitation facilities was associated with lower risk of ARI (OR = 0.55, 95% CI 0.45 to 0.68), diarrhoea (OR = 0.83, 95% CI 0.71 to 0.96), and stunting (OR = 0.64, 95% CI 0.49 to 0.86). Investments targeting synergies in integrated energy and water, sanitation and hygiene, and women empowerment programmes are likely to contribute to the reduction of the burden from early childhood illnesses. Research and development actions in LMICs should address and include multi-sector synergies.

## 1. Introduction 

Household microenvironment determinants including household air pollution, unsafe drinking water and sanitation, and poor hygiene have detrimental health impacts [1,2,3]. Despite progress made during the last two decades to address these health risks, lack of access to clean energy, water, and sanitation remain a challenge in low- and middle-income countries (LMIC) [4]. Cook stoves using polluting fuels are widespread in LMICs with serious health implications [5]. Around 3.1 billion people in LMICs rely on polluting energy sources for cooking as more than 95% of all households rely primarily on biomass for cooking in Africa [6]. It is projected that by 2030 about 600 million people will still be without access to electricity in sub-Saharan Africa (SSA) [7]. In 2015, during the end of the millennium development goals, 844 million people were expected to still lack basic drinking water service and 2.3 billion people still lacked basic sanitation service, mainly in SSA [8]. These trends in access to clean energy as well as to water and sanitation are major household microenvironment risks that are in the goals as formulated in the UN Sustainable Development Goals (SDG), presently setting the international and national policy agendas. Likewise, women empowerment has reached the top of agendas of the SDG.

Household environment risk factors are among the leading causes of death worldwide, mostly in LMICs. Globally in 2016, 3.8 million deaths were attributable to household air pollution, almost all in LMICs. In the same year, unsafe sanitation and hygiene were responsible for nearly 0.9 million deaths. Africa disproportionately suffered the burden from such deaths [9,10], and the number of people dying from water-pollution-related complications is highest in SSA [11]. In particular, young children in LMICs are highly vulnerable to household-environment-related risk factors [6,10,12,13,14]. By 2030, environmental health risk factors, including undernutrition and malaria, will still be responsible for 14% to 16% of total global child deaths. 

Polluting energy sources and cooking technologies are among the leading contributors to the global burden of diseases [11,15]. The most important disease associated with the use of household polluting energy sources s is acute respiratory infection (ARI) in children [12,16,17,18,19,20]. Studies show that polluting energy sources, in particular biomass fuel, are linked with the prevalence of stunting in children [13,21,22]. Major adverse health outcomes in children linked with unsafe water sources, inadequate sanitation, and hygiene include diarrhoea [17,23,24,25,26], stunting [23,25,27], and respiratory infection [11]. Other studies show associations between women’s empowerment and stunting and mortality in children under 5 years of age and poverty reduction [28,29,30,31,32]. 

Empirical evidence on the association between household microenvironments, women empowerment, and health outcomes of young children is important to guide policies. Previous studies produced informative empirical evidence on the topic, but most of them focused on one or a couple of household environment risk factors [16,23,24,27], while others focused on women empowerment [28,31,33,34,35]. A recent study [36] more comprehensively assessed determinants of child health outcomes in Africa but used aggregate country level environmental exposure data. In general, comprehensive analyses addressing the interplay between household level microenvironment risks, women empowerment, and health outcomes of children in LMICs, especially in SSA, are limited. Previous studies focused on the association between water, sanitation, and hygiene and child growth in Eastern Africa [37] and with diarrhoea in Uganda [38]. A comprehensive understanding of the association between household environmental risk factors and health outcomes of children is lacking in Uganda.

In this study, we aim to extend the understanding of the association between household microenvironments and women empowerment and health outcomes of children under five years of age. We make use of recent methodological developments in indexing household energy poverty, as a proxy for household air pollution, and the women empowerment index. The multidimensional energy poverty is estimated to quantify households clean energy deprivation following Nussbaumer, et al. [39] that employs capability theory [40]. Likewise, we estimated a survey-based women empowerment index [41] and used it to measure and include women empowerment. 

## 2. Materials and Methods

In Figure 1, we present a simplified conceptual framework to show possible links between energy poverty, access to clean water, sanitation, hygiene, women empowerment, and health outcomes of children. Our conceptual approach incorporated long-standing accepted linkages between household microenvironments, women empowerment, and health outcomes of children. We also incorporated recently developed indices for environmental exposure [39] and women empowerment [41] to have a better understanding of the context-specific relationship between the exposures and health outcomes. The women empowerment index is useful to identify what types of empowerment are linked with health outcomes of children.

### 2.1. Data Source 

The data analysed in this study are from the 2016 Uganda Demographic and Health Surveys (UDHS), collected by the Uganda Bureau of Statistics with technical support from ICF international [42]. The UDHS is a nationally representative survey that provides comprehensive data about households, health outcomes of children, and maternal characteristics. The survey was carried out from 20 June to 16 December 2016 on key demographic and health indicators, including nutritional status of children and women and gender-related variables. A stratified and multistage sampling method was used in the 2016 UDHS to collect key information on child and maternal health indicators, which is nationally representative. A detailed description of methods, design, collected data, study participants and other important information is documented in the 2016 Demographic and Health Survey of Uganda [42]. The survey has rich data including information about children’s morbidity, though most of them are symptomatic data. It also has rich data on household and parents’ socioeconomic and demographic characteristics. Therefore, the UDHS survey data are attractive for rigorous quantitative analysis to establish an association between morbidity among children under 5 years of age and household microenvironments and women empowerment. 

### 2.2. Description of Health Outcomes and Predictors 

#### 2.2.1. Health Outcomes 

The key health outcome variables in this study came from the mother’s responses to questions on episodes of various child morbidity within two weeks before the survey date. As indicated in Figure 1, the child health outcomes in this study are ARI, diarrhoea, and nutritional status of children using stunting as a key indicator. In the 2016 DHS surveys, symptoms of ARI are defined as short, rapid breathing which was chest-related and/or difficult breathing which was chest-related [43]. Following this definition, the children were categorised into two groups: those who experienced ARI symptoms and those who did not within 2 weeks before the survey. A limitation of this indicator is that it is based on the mothers’ perception of the morbidity, not a definitive diagnosis. The DHS data also contains diarrhoea prevalence by asking mothers whether the child had diarrhoea in the two weeks preceding the survey. This health outcome is dichotomous, identifying children who suffered diarrhoea and those who did not. 

Stunting in children below 5 years of age is the other health outcome examined in this study. Following the WHO Multicentre Growth Reference guideline [44,45], a child is stunted if the height-for-age z-score (HAZ) is below -2SD of the median for their age, including both mildly and severely stunted children.

#### 2.2.2. Independent Variables 

Key predictors of interest in this study are a comprehensive set of variables related to household microenvironments and women empowerment, as presented in Figure 1. Most household-environment-related variables in the DHS are standardized in the recode files and often used as they are with moderate modifications; for example, see [46]. In this study, however, indices were constructed and globally set standards were used to categorize key variables of interest. The multidimensional energy poverty index [39] was constructed and used as an indicator for household air pollution. We also estimated an women empowerment index, relevant in African settings. To include water quality and sanitation and hygiene facilities, we used the revised standard ladder by WHO/UNICEF Joint Monitoring Program [8]. We discuss these measurements and standards in the following sections. 

#### 2.2.3. Multidimensional Energy Poverty Index (MEPI)

An indicator for household air pollution is a key predictor in this study. Previous studies often considered households’ consumption of solid fuel as an indicator of household air pollution to explain childhood morbidity [18,21,47,48,49,50]. In this study, we constructed a multidimensional energy poverty index (MEPI) at the household level to explain childhood morbidity. The MEPI captures a set of energy deprivations that affect a person or household [39,51]. The MEPI provides a framework to identify the categories of households left behind on access to clean, safe, and sustainable household energy [52]. Methodologically, the MEPI is derived from the multidimensional poverty measures developed by Alkire and Foster [53]. Literature on methodological developments of multidimensional poverty have their root in Amartya Sen’s discussion of deprivations and capabilities [54] which argues for the need to focus on the absence of opportunities and choices for living a basic human life. The MEPI is composed of five dimensions representing basic energy services with six indicators. More specifically, it is composed of indicators of a household using modern cooking fuel and cooking places, having access to electricity for lighting, having a refrigerator, having a TV or radio for entertainment and education, and having a phone or mobile phone for communication (see Table 1). 

Following previous studies [39,51,52] and the relative importance of indicators to human health, we unequally assigned weights to the various dimensions and indicators. This reflects the relative importance of the various energy poverty variables considered in household pollution and human health. We refer readers to Nussbaumer, Bazilian, and Modi [39] for further understanding of dimensions, indicators, and weights used in MEPI construction. 

A household is identified as energy poor if the respective set of deprivation scores $\left(C_{i}\right)$ exceeds a predefined threshold, k. Previous studies in LMICs used a multidimensional energy poverty cut-off score at k=0.3 [39,52]. Nussbaumer, Nerini, Onyeji, and Howells [51] further categorized a household multidimensional energy poor level as acute when the MEPI exceed 0.7, moderate between 0.3 and 0.7, and low below 0.3. We also followed these cut-off points. However, in our analysis, very few (0.16%) of the observations fell in the low multidimensional energy poor category. Consequently, we combined the ‘low’ and ‘moderate’ energy poor category and coded as ‘moderate’ multidimensional energy poor. Therefore, households were categorised into ‘moderate’ and ‘acute’ multidimensional energy poor. 

The MEPI captures information on both the incidence and the intensity or severity of energy poverty. We computed the poverty headcount as H=qn, where q is the number of energy poor households (where ci>k) and n the total number of households. The severity of poverty indicates the average proportion of indicators in which multidimensional energy poor households is obtained as $A=\overset{n}{\underset{i=1}{\sum }}ci\;\left(k\right)/q$. Finally, the MEPI is obtained as the product of the multidimensional energy poverty headcount ratio (H)  and multidimensional energy poverty intensity (A): MEPI=H×A. 

#### 2.2.4. Water, Sanitation, and Hygiene Facilities 

In coding quality and source of household water and sanitation, previous studies used the dichotomous improved vs. unimproved or safe vs. unsafe definitions [17,48,55]. As our focus in this study is household microenvironments, we opted for definitions and categorization that are clearer and more distinct. We followed the revised water and sanitation ladder by WHO/UNICEF joint monitoring programme (JMP) [8] to define quality of water and sanitation facilities but with some modifications. We defined household access to a hygiene facility following the WHO/UNICEF joint monitoring programme ladder for hygiene: basic, limited, and no facility. The WHO/UNICEF defines and categorizes quality of water and sanitation facilities into five levels. This was not possible in the Uganda DHS dataset due to less clarity in wording used in the questionnaire to match with WHO/UNICEF JMP ladders for water and sanitation. We categorised sanitation facilities, slightly deviating from the WHO/UNICEF JMP ladder for sanitation, into three: no facility, unimproved, and improved. We categorized the quality of the source of drinking water into two: improved and unimproved. 

#### 2.2.5. Women Empowerment Index

We adopted the women empowerment index construction method developed by [41] using DHS data from Africa. This composite index consists of three domains of empowerment: attitude to violence, social independence, and decision making. These empowerment dimensions overlaps with most of the dimensions considered by [56] focusing on sub-Saharan Africa, particularly East African countries. Similar empowerment dimensions were considered in other studies in LMICs [29]. The three empowerment dimensions comprise various information. ‘Attitude to violence’ is composed of information related to the respondent’s opinion about whether wife-beating is justified or not in various scenarios. ‘Social independence’ includes items related to education, frequency of information consumption (reading), age at first cohabitation and first childbirth, and differences between age and their years of schooling of the woman and her husband. The ‘decision making’ domain is comprised of information related to a woman’s involvement in household decisions and labour force participation. In this study, the three dimensions were weighed following [41].

In addition to the household microenvironments and women empowerment-related predictors, other relevant predictors such as individual and parental variables are included in the analysis. Potential predictors associated with the health of children below 5 years of age were included, considering their relevance in previous studies [16,21,23,25,29,30,46,47,57,58] as control variables. A summary of definitions of key predictors used in the analysis is presented in (Table 2).

### 2.3. Statistical Analyses 

The statistical analyses used in this study include descriptive statistics, bivariate analyses, and logistic regression models. The DHS sampling weights are applied in all analyses to account for the complex survey design. We used descriptive analyses, percentages, and numbers to show the distribution of childhood morbidity and nutritional status by predictor variables. Associations between childhood health outcomes and predictors were first analysed using bivariate analyses, the $\chi^{2}$ tests, before fitting the regression models. These analyses were carried out to compare the prevalence of childhood morbidities and stunting among the levels of the selected predictors and to inform further analyses using regression models. 

The dependent variables considered in this study are dichotomous variables. Therefore, binary response econometric models are the natural choice. Logistic regression models were estimated to evaluate the association between key predictors and health outcomes of children considered for this study. The logistic regression model we estimate as:$$ln\left(p_{ij}/{\left(1-p\right)}_{ij}\right)={Χ}^{\prime }\beta$$
where $p_{ij}$ is a dichotomous health outcome for child i in household j, β denotes vector of coefficients estimated, Χ denotes a set of values of predictors: household energy poverty, water, sanitation and hygiene, women empowerment, and control variables. We used the Stata 15 software package for all data analyses and logistic regressions, and we reported odds ratios. 

## 3. Results 

Table 3 shows the results from analyses using descriptive statistics. These results shows that about 77% of children lived in households with access to an improved source of drinking water, but still a sizable proportion of the children lived in households that use unimproved sources of water, including surface water as primary drinking water sources. A large percentage of children lived in households with access to sanitation facility, but most of them (59%) were unimproved facilities. The lack of access to a hygiene facility is more pronounced, and the results indicate that 42% of households had no hygiene facility. Descriptive statistics analysis, based on a multidimensional energy poverty index (MEPI), shows that energy poverty is widespread in Uganda. About 26% of children lived in households who were in acute multidimensional energy poverty, while 64% of them lived in households that were in moderate energy poverty. This suggests that most households did not use a clean energy source or did not benefit from energy services supplied by electricity. Close to 80% of the children live in a rural area, and the average family size was about six persons, a fairly consistent result in sub-Saharan Africa (see also [16]). A larger percentage of children were from parents with a primary level of education (62% of mothers and 55% of fathers). The percentage of mothers with no education was larger than fathers, while the percentage of fathers with a secondary and above level of education was higher than mothers. 

Results shown in Table 3 reveal that more than 9% of the children under five years of age in Uganda experienced an episode of ARI, two weeks before the survey. This is lower than overall ARI prevalence in LMICs using older DHS data, phase-V DHS survey [58], and also lower than ARI prevalence in Bangladesh using recent DHS data [47]. The result about growth retardation shows that 30% of under-five children in Uganda in the DHS sample have stunted growth. About 20% of children experienced an episode of childhood diarrhoea, in the two weeks prior to the DHS 2016 survey. 

Table 4 shows the distribution of the prevalence of episodes of childhood morbidities and stunting together with the associated $\chi^{2}$ tests. Incidences of ARI and diarrhoea were significantly likely among children below three years of age, but stunting was significantly more prevalent among children above one year of age, more so among 12–35 months of age. The sex of children showed significant association with episodes of diarrhoea and stunted growth. The educational level of mothers showed a significant association with the prevalence of the episodes of ARI and stunting, but not with diarrhoea in children in Uganda. The results in Table 4 also reveal that the episodes of childhood morbidity and stunting are significantly associated with household-environment-related covariates. The prevalence of incidences of diarrhoea is likely more prevalent among children from households with a poor-quality source of drinking water and sanitation and hygiene facilities. Childhood stunting is more likely among children from households with access to an unimproved source of drinking water. The household multidimensionally energy poverty is significantly associated with ARI and stunting in children (*p* < 0.01) and diarrhoea (*p* < 0.05). Socioeconomic status and residence area, rural or urban, of households also is significantly associated with health outcomes in children. 

Associations of household energy poverty, type of drinking water sources, access to sanitation and hygiene facilities, women empowerment, and children’s health outcomes were analysed in detail using econometric models. Three logit model specifications were used to analyse the association between MEPI and ARI. The first model controlled for individual covariates only. The second model controlled for individual covariates, type of drinking water sources, and access to sanitation and hygiene facilities. The third model controlled for all covariates included in model two and women empowerment. Results from the three model estimations are given in the Appendix A. The results from the three estimated models showed consistent association between MEPI and ARI. Therefore, the third model which controlled for individual covariates, household microenvironments-related covariates, and included women empowerment is reported here. 

The association of household microenvironment, women empowerment, and child-related predictors with childhood morbidities and stunting is presented in Table 5. This result shows odds ratios (ORs) and the associated confidence intervals (CI) for predictors included in the analysis. After controlling for all other variables, the odds of experiencing ARI and diarrhoea were significantly higher among younger children. Children above three years were about 33% less likely to experience ARI, compared with children below one years old, but there was no significant difference between children under one year and children who were one to three years old. Compared with children below one year, older children were 21% to 75% less likely to experience diarrhoea. In contrast, older children, between one and five years of age, were 154% to 265% more likely to experience stunting compared with children below one year old. Children from rural areas of Uganda were 14% more likely to experience diarrhoea compared with children from urban areas. 

The results suggest that after controlling for other variables, multidimensional energy poverty and sanitation and hygiene facilities are significantly associated with the prevalence of ARI among young children in Uganda. Once all other potential predictors are controlled for, children from households that are energy poor were 32% more likely to experience an episode of ARI, compared with children from moderate energy poor households. Our result also implies access to sanitation facilities is a better preventive measure for provenance of ARI. Children belonging to households with access to any kind of sanitation facilities, compared with children in households without sanitation facilities, were about 45% less likely to experience an incidence of ARI. Children in households with access to basic quality hygiene facility had 25% lower odds of suffering from ARI, and the reduction in an ARI episode for a limited quality hygiene facility is about half of the basic hygiene facility. Women’s disagreement with wife-beating, the attitude to the violence domain of women empowerment (OR: 0.88, 95% CI:0.82 to 0.93) was significantly inversely associated with ARI prevalence. The social independence domain (OR: 0.91, 95%CI: 0.84 to 0.98) is also significantly associated with the incidence of ARI, but the reduction in odds of prevalence was modest. 

Households’ sanitation and hygiene facilities are significantly associated with episodes of diarrhoea among children, controlling for other variables. Children in households with access to basic and limited hygiene facility had 20% and 16% lower odds of experiencing diarrhoea, respectively, compared with children from households without hygiene facility in their premises. Though unimproved, access to toilet facilities was protective of the incidence of diarrhoea in children compared with households without a toilet facility. It reduces the likelihood of an episode of diarrhoea by 17%. A more interesting result in the analysis is significant association between domains of women empowerment and episodes of diarrhoea in children, implying the need to measure, monitor, and improve women empowerment for better health outcomes of children. Among the three domains of women empowerment, the attitude to violence (OR: 0.89, CI: 0.84 to 0.93) and social independence (OR: 0.93, CI (0.88 to 0.99) domains were significantly associated with the incidence of diarrhoea in children. These two domains are more about if women believe violence against women is justifiable and women’s access to information, educational, and age difference with their spouses. Therefore, women’s access to education, information, and freedom from violence are still intervention areas for better health outcomes of children. 

Result from the analysis of predictors of stunting revealed that access to improved water and sanitation and women empowerment are significantly associated with stunting, controlling for all other variables. Children from households with access to improved source of drinking water and sanitation are about 17% and 36%, respectively, less likely to have stunted childhood growth. Social independence of women (OR: 0.83, CI:0.75 to 0.92) was also significantly associated with stunting. Child and mothers related predictors considered in this analysis are also significantly associated with stunting in children. 

## 4. Discussion

A main finding of our study shows that multidimensional energy poverty, an indicator for household deprivation from access to clean energy sources, was a strong predictor of the prevalence of ARI among young children in Uganda. This is a first among the studies of this kind. Studies in the past used various proxies for households’ clean energy access and its effect on health outcomes of children [18,22,47]. Compared to other proxies for households’ energy sources, this finding shows to be consistent with previous studies on the association of households’ energy sources and ARI in LMICs [17,49,50]. This reaffirms the critical role of improving households’ access to clean energy sources to address the burden of respiratory diseases in LMICs. This is particularly critical as children are vulnerable to polluting household environment in LMICs.

The second important finding is the strong and consistent association between women empowerment and child health outcomes. This finding is consistent with previous studies on women empowerment and health outcomes of children in LMICs [29,31,59]. We showed that women empowerment domains, particularly freedom from domestic violence and social independence of women, were vital to reduce the prevalence of ARI, diarrhoea, and stunting among children. The social independence domain of women empowerment is predominantly about women’s level of education, access and/or processing of information. This implies that it is important to improve women’ access to education and information so that they can make choices and decisions that affect health outcomes of their children in a positive way. Therefore, our findings underscore the importance of education to achieve the twin goals of women empowerment and a reduced burden of childhood illness.

In addition to these findings related to ARI, in the bivariate analysis, diarrhoea and stunting were significantly associated with household energy poverty. In the multivariate analysis, however, the result shows no significant association between multidimensional energy poverty and diarrhoea and stunting in children. This is similar to previous study by Machisa, Wichmann, and Nyasulu [22] who used biomass fuel consumption as a proxy for polluting household energy sources. Consistent with previous studies in LMICs [23,25,48], households’ access to improved sources of drinking water was preventive of childhood stunting. 

Another finding on household microenvironments and health outcomes of children is the preventive effect of household sanitation facilities, affirming the findings in other studies and contributing to the internal and external validity of our analyses [23,24,36]. Our result strengthens the evidence that households’ access to sanitation facilities are critical to reduce ARI, diarrhoea, and stunting in children.

Some findings are different from the findings of other studies. Handwashing is a common finding as inversely associated to the incidence of diarrhoea and transmission of influenza [60,61] in LMICs. Our study shows that hygiene was an important predictor of ARI and diarrhoea, but not stunting. ARI and diarrhoea in children are more prevalent among younger children. Hasan and Richardson [17] also reported a similar result using DHS data from Bangladesh, Nepal, and Pakistan. However, stunting was significantly more prevalent in the group of older children among those under five years of age. Our observation is consistent with a study from Nigeria [55]. One would expect a different and perhaps changing causal dynamic over time in the development of stunting. 

## 5. Conclusions 

In this study, we measured the associations of household microenvironment risks and health outcomes of young children, focusing on multidimensional energy poverty and women empowerment. We used the largest, nationally representative available health and demographic dataset. In our very comprehensive approach, we identified household microenvironment risk factors and women-empowerment-related factors associated with ARI, diarrhoea, and stunting among young children. 

In our analyses, household microenvironments and women empowerment showed a strong association with childhood health outcomes, though with varying magnitudes of associations. ARI is particularly associated with household multidimensional energy poverty, women empowerment, sanitation, and hygiene. Consistent findings were obtained using both bivariate and multivariate analysis. One of the key contributions to the literature in this study is the use of a Multidimensional Energy Poverty Index (MEPI) as a proxy for pollution in the household. Our findings are consistent with previous studies that used other proxies for household pollution. One of the dimensions of MEPI is households’ deprivation from clean energy sources for communications and information. This has important implications for women empowerment, particularly the social independency dimension of empowerment. Therefore, targeting complementarity of clean energy access and women empowerment is important to address the burden of early childhood illness. 

Access to sanitation and hygiene facilities and women empowerment are important targets for intervention to reduce diarrhoea and stunting in children. Investments targeting synergies in integrated interventions targeting household microenvironments and women empowerment are important. Research and development efforts in these areas are criticized for over specialization and lack of an integrated approach [62]. We argue for research and development activities in LMICs needing to target the potential synergies and complementarities among clean energy access, women empowerment, sanitation, and hygiene to achieve the UN 2030 goals in areas of health and development.

Our new evidence on health risks associated with energy poverty suggests a need for a continued effort in accelerating access to clean energy for poor households (main goal under SDG-7). However, evidence shows that the world is progressing slowly in achieving SDG-7 [6], undermining the observed gains in SDG-3 (child health). This slow progress potentially undermines gains in women empowerment (main goal under SDG-2 and -5) due to women’s dominant domestic role. The association between households’ access to clean water, sanitation, and hygiene (priorities under SDG-6) and childhood health outcomes also suggests a need for a renewed effort to progress towards achieving SDG-6 to realize the potential gains in SDG-3. Hence, it is important to target the potential multi-sector synergies in household microenvironment improvements to achieve SDG-3. 

Based on our two key findings, we provide new evidence that supports global and national efforts to achieve SDGs. Specifically, in (child) health these efforts should target the potentialmultisectoral synergies and complementarities of household-level programmes. The research community could play important roles in evaluating the synergies and building an empirical evidence base for better use of resources to achieve better childhood health. 

## 6. Limitations 

In this comprehensive study, we used the largest available health and demographic dataset. We used standard definitions and categorizations of household water, sanitation, and hygiene. Yet we recognize caveats pertaining to measurements and survey methodologies, in addition to the limitations of cross-sectional surveys in general, especially the lack of a time dimension and the drop-out of children in the households who already died. 

Episodes of ARI and diarrhoea are based on women’s self-reports, not based on objectively measured results from clinical examinations. In particular, ARI definition is based on combinations of symptoms reported in the mothers’ questionnaire. This could induce potential measurement error and bias the results. The bias is minimalized in case of a short recall period. Two weeks prior to the survey was used to collect data on episodes of ARI and diarrhoea. Due to our interest in women empowerment in the analysis, only children whose mother was interviewed were considered in the analysis. The exclusion may lead to bias to the health outcomes considered in this study. However, the number of observations is large; hence, the magnitude of bias is probably small. 

We consider the generated evidence about women empowerment and child health outcomes as robust, outweighing the loss in observations from the survey. The household microenvironment, women empowerment, and health outcomes are expected to be changing over the time since 2016 in Uganda, yet the evidence about associations between predictors and outcomes is still relevant and generalizable for many settings. 

## Figures and Tables

**Figure 1 ijerph-19-06684-f001:**
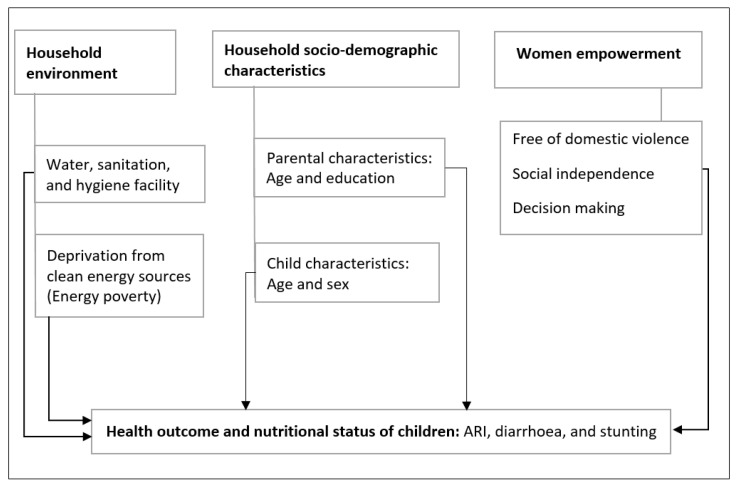
Conceptual framework: household environment (including social dimension) and health outcomes of children below 5 years of age. The bolder lines show key associations that we aim to test while the lighter lines show control variables.

**Table 1 ijerph-19-06684-t001:** Multidimensional energy poverty dimensions and respective variables with cut-offs, including relative weights (in parenthesis).

Dimensions	Indicator (*Weight)*	Variables	Deprivation Cut-Off (Energy Poor if…)
Cooking	Modern Cooking fuel *(0.2)*	Type of cooking fuel	any fuel use besides electricity, LPG, kerosene, natural gas, or biogas
	Indoor pollution *(0.2)*	Food cooked on stove or open fire (no hood/chimney), indoor, if using any fuel besides electricity, LPG, natural gas or biogas	True
Lighting	Electricity access *(0.2)*	Has access to electricity	False
Services provided by means of household appliances	Household appliance ownership (*0.13)*	Has a refrigerator	False
Entertainment/education	Entertainment/education appliance ownership *(0.13)*	Has a radio OR television	False
Communication	Telecommunication means *(0.13)*	Has a phone landline OR mobile phone	False

Source: taken from Nussbaumer, Bazilian, and Modi [39].

**Table 2 ijerph-19-06684-t002:** Description of key predictors used in the analysis.

Variables	Descriptions
**Multidimensionally energy poverty**	Categorized into acute (MEPI exceed 0.7) and moderate (MEPI ≤ 0.7)
**Women empowerment**	Scores of the empowerment dimensions (attitude to domestic violence, social independence, and decision making) estimated from components of dimensions and their weight
**Source of drinking water**	Categorized into improved and unimproved
Unimproved	Drinking water directly from a river, dam, lake, pond, stream, canal, or irrigation canal and drinking water from an unprotected dug well or unprotected spring
Improved	Drinking water located in the premises or not, from piped water, protected dug wells, protected springs, rainwater, and packaged water. This category also includes drinking water from boreholes or tube wells.
**Sanitation facility**	Categorized into poor, improved, and safely managed
No facility	No toilet facility
Unimproved	Use of pit latrines without a slab or platform, hanging latrines or bucket latrines
Improved	Sanitation facilities including flush/pour flush to piped sewer systems, septic tanks, or pit latrines; ventilated improved pit latrines, composting toilets or pit latrines with slabs
**Hygiene facility**	Categorised as no facility, limited, and basic
No facility	No handwashing facility on premises
Limited	Availability of a handwashing facility on premises without soap and water
Basic	Availability of a handwashing facility on premises with soap and water

**Table 3 ijerph-19-06684-t003:** Descriptive characteristics of study participants, *n* (percentage) or mean (SD).

Characteristics	*n* (Percentage) or Mean (SD)
**Child demographic characteristics, *n* (%)**	
Child is female	7326 (49.8)
Child age is 0–11 months	2622 (21.5)
Child age is 12–35 months	4796 (39.3)
Child age is 36–59 months	4795 (39.3)
**Parental characteristics**	
Mother’s age, mean (SD)	28.42 (6.75)
Mother no education, *n* (%)	1937 (13.17)
Mother Primary education, *n* (%)	9187 (62.45)
Mother Secondary or higher education, *n* (%)	3586 (24.38)
Partner’s age, mean (SD)	44.92 (24.8)
Husband no education, *n* (%)	837 (7.03)
Husband Primary education, *n* (%)	6572 (55.21)
Husband Secondary or higher education, *n* (%)	4495 (37.76)
**Household characteristics**	
Rural resident	11,398 (78.65)
Household size, mean (SD)	6.18 (2.75)
Children under 5 in the household, mean (SD)	1.91 (0.94)
**Source of drinking Water, *n* (%)**	
Unimproved	3305 (22.81)
Improved	11,187 (77.19)
**Sanitation facility, *n* (%)**	
No toilet facility	1031 (7.11)
Unimproved	8498 (58.64)
Improved	4963 (34.25)
**Hygiene facility, *n* (%)**	
No facility	6137 (42.34)
Limited	4915 (33.92)
Basic	3441 (23.74)
**Household multidimensional energy poverty, *n* (%)**	
Multidimensionally acute energy poverty	3822 (26.39)
Moderate energy poverty	10,660 (73.61)
**Wealth Index, *n* (%)**	
Poorest	3251 (22.43)
Poorer	3038 (20.96)
Middle	2799 (17.31)
Richer	2579 (17.79)
Richest	2826 (19.50)
**Child health outcomes, *n* (%)**	
ARI	1354 (9.34)
Diarrhoea	2832 (19.54)
Stunted	1479 (28.9)

**Table 4 ijerph-19-06684-t004:** Sociodemographic characteristics by health outcomes of children along with the *χ*^2^ tests of association.

Characteristics	Child Health Outcome, (%)		
	ARI	Diarrhoea	Stunted
**Age of child**			
0–11 months	10.71	29.43	14.10
12–35 months	10.65	24.44	36.26
36–59 months	7.63	9.36	27.71
*χ* ^2^	43.90 ***	722.61 ***	180.41 ***
**Sex of child**			
Female	9.10	18.09	26.88
Male	9.70	20.99	30.90
*χ* ^2^	2.28	20.85 ***	7.44 **
**Mother’s education**			
No education	11.81	18.62	37.81
Primary	9.60	19.95	30.10
Secondary or higher	7.83	19.00	19.82
*χ* ^2^	24.31 ***	2.71	81.74 ***
**Household drinking water quality**			
Unimproved	9.97	19.2	32.76
Improved	9.16	19.64	26.79
*χ* ^2^	2.1	0.34	16.34 ***
**Household Sanitation facility**			
No facility	15.75	23.29	33.63
Unimproved facility	9.27	19.78	31.61
Improved facility	8.15	18.35	21.53
*χ* ^2^	62.38 ***	14.70 **	64.39 ***
**Household Hygiene facility**			
No facility	11.15	22.21	28.01
Limited	8.78	17.94	29.98
Basic	6.91	17.05	26.18
*χ* ^2^	53.01 ***	52.99 ***	5.63
**Multidimensional energy poverty**			
Moderate energy poverty	6.96	17.46	23.85
Acute energy poverty	10.38	20.36	29.95
*χ* ^2^	50.71 ***	18.16 **	22.10 ***
**Residence**			
Rural	7.07	20.23	29.63
Urban	9.96	16.99	22.39
*χ* ^2^	25.78 ***	17.37 ***	21.93 ***
**Wealth Index**			
Poorest	12.74	22.18	32.51
Poorer	10.46	21.32	32.31
Middle	8.86	18.7	31.59
Richer	8.31	18.47	26.56
Richest	5.9	16.55	16.63
*χ* ^2^	108.32 ***	42.53 ***	93.98 ***

*** *p* < 0.01, ** *p* < 0.05.

**Table 5 ijerph-19-06684-t005:** The odds ratios and confidence intervals with significance level for the studied covariates from the logistic regression models for ARI, diarrhoea, and stunting.

Variables (Base Category)	Odds Ratios (ORs) and 95% Confidence Intervals (CI)		
	ARI	Diarrhoea	Stunted
**Multidimensional energy poverty (moderate)**			
Multidimensionally Acute Energy Poor	1.32 *** (1.10 to 1.58)	0.98 (0.86 to 1.11)	1.12 (0.92 to 1.37)
**Drinking water quality (unimproved)**			
Improved	0.99 (0.85 to 1.5)	1.08 (0.97 to 1.20)	0.83 ** (0.70 to 0.99)
**Sanitation facility (No facility)**			
Unimproved	0.55 *** (0.47 to 0.65)	0.83 ** (0.71 to 0.96)	0.89 (0.70 to 1.14)
Improved	0.55 *** (0.45 to 0.68)	0.91 (0.76 to 1.08)	0.64 *** (0.49 to 0.86)
**Hygiene facility (No facility)**			
Limited	0.87 * (0.77 to 1.01)	0.84 *** (0.75 to 0.93)	1.05 (0.88 to 1.25)
Basic	0.75 *** (0.63 to 0.89)	0.80 *** (0.70 to 0.91)	1.01 (0.83 to 1.23)
**Scores for the three women empowerment domains**			
Score for attitude to violence	0.88 *** (0.82 to 0.93)	0.89 *** (0.84 to 0.93)	1.06 (0.97 to 1.16)
Score for social independence	0.91 ** (0.84 to 0.98)	0.93 ** (0.88 to 0.99)	0.83 *** (0.75 to 0.92)
Score for decision making	1.05 (0.98 to 1.12)	1.00 (0.95 to 1.05)	0.95 (0.87 to 1.03)
**Sex of child (Male)**			
Female	0.93 (0.82 to 1.04)	0.86 *** (0.79 to 0.94)	0.78 *** (0.68 to 0.91)
**Child age category (0–11** **months)**			
12–35 months	1.001 (0.86 to 1.17)	0.79 *** (0.71 to 0.88)	3.65 *** (2.90 to 4.61)
36–59 months	0.67 *** (0.57 to 0.79)	0.25 *** (0.22 to 0.29)	2.54 *** (2.00 to 3.23)
**Place of residence (Urban)**			
Rural	1.12 (0.92 to 1.36)	1.14 * (0.99 to 1.32)	1.10 (0.86 to 1.42)
Mother’s height in meter (<14.5)			
≥14.5			0.28 *** (0.21 to 0.38)
**Mother’s BMI (<18.5 Kg/m^2^)**			
18.5 to 24.9 Kg/m^2^			0.87 (0.67 to 1.12)
≥25 Kg/m^2^			0.61 *** (0.45 to 0.82)
**Child’s birth order (first)**			
Second			1.09 (0.84 to 1.41)
Third			1.32 ** (1.02 to 1.72)
Fourth and above			1.02 (0.82 to 1.28)
Constant	0.17 *** (0.13 to 0.24)	0.49 (0.38 to 0.63)	0.73 (0.42 to 1.30)
Number of observations	12,095	12,095	3694
$\chi_{df}^{2}\;\left(P-value\right)$	203.81 (*p* < 0.01)	572.15 (*p* < 0.01)	263.12 (*p* < 0.01)
McFadden’s or the Pseudo *R*^2^	0.025	0.06	0.07

*** *p* < 0.01, ** *p* < 0.05, * *p* < 0.1.

## Data Availability

Data are available upon request from the Demographic and Health Survey Program (https://dhsprogram.com/Data/).

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
