# Peer review of "Household Microenvironment and Under-Fives Health Outcomes in Uganda: Focusing on Multidimensional Energy Poverty and Women Empowerment Indices"

_ijerph, 2022, doi:10.3390/ijerph19116684_

Round 1
Reviewer 1 Report
(Please see the attached page for detailed comments).
In general I am pleased to read a manuscript for this Journal that does not need extensive re-writing due to difficulties with the English language. In fact, this one was a pleasure to read, being extremely well written and employing appropriate methods.
The work also represents a further expansion into the realm of social determinants of health within a public health enquiry.

Author Response
Reviewer 1
We are grateful to the reviewer for taking their time to read the manuscript. We received invaluable comments and feedback from the reviewer. In the following lines, we try to highlight how we addressed all comments raised by the reviewer.
Note: our responses are written in blue.
Reviewer: Section 2.2.1 “... WHO Multicentre Growth Reference guideline [37,38], A child is stunted if height for-age z-score (HAZ) is below -2SD of the median for their age...” Such standard measures often run afoul of a distribution that is other than near-normal, in which case the 2SD cut off is unrealistic or meaningless! Can readers be assured that distributions were checked for normality?
Authors : Thank you, this is important. We followed the well-established method, the widely used in the DHS datasets. We have rechecked normality of the HAZ. It has very near normal distribution apart from couple of outliers that have been flagged as invalid data. A
Reviewer: It seems that MANY of the variables used are inter-related, suggesting the potential for interactions leading to confounding or effect-modifying. Is this adequately addresses?
Authors: we agree this is also important point. We have made an effort to address this in the initial submission. Variables that seem inter-related are the exposure variables. We run different models including only one exposure, then including one more exposure, finally the full model including all exposures. As you have seen the reported result shows all different model specifications gave consistent result. Following your comment, we have tried to test if confounding is an issue using the ‘chest’ Stata command. Our test results suggest that the confounding effects would not alter the conclusion.
Reviewer: Section 2.2.5 Most in-text citations in the manuscript are suitably abbreviated in agreement with standard style guides. But ONE citation is unusually fully-displayed with all six authors: “ ... Ewerling, Lynch, Victora, van Eerdewijk, Tyszler and Barros [34] ...”. Further, this citation appears FOUR times in a single paragraph! The reader is left wondering is this is a form of highprofile “product placement” for an academic audience! Perhaps, a more uniform approach could be adopted with regard to abbreviating in-text citations.
Authors: This is correct. Thank you for carefully reading it. We entirely relied on the journal’s from Endnote. We have now corrected it.
Reviewer: Table 3.... line 3: Child spelled incorrectly
Authors: this now corrected.
Reviewer: Table 4 and Table 5. The legend for Table 4 shows: *** P<0.01 ** P<05. This is adequate, but in Table 5, no such legend is shown. Is the same interpretation to be made? What about * ? There are several results showing * . Is this a level of statistical significance other than P<05. It is an important question, because There are examples of 95% confidence limits including 1.0, where a single * is displayed. 0.87* (0.77 to 1.01) Can this be addressed/clarified?
Authors: This has been checked throughout the text We have included legend in table 5 too. We have also included the legend for *. The * stands for p<0.1. We understand it is not very common to report P<0.1, but this work is a multidisciplinary research and different disciplines have different traditions of reporting. Hope it now clear.
Reviewer 2 Report
Very interesting and relevant study, with high possibilities to be replicated in other parts of the world.
My only concern is that MEPI does not include any energy services, or considerations to energy deprivations, related to thermal comfort. I think that acknowledging the importance of thermal comfort as a public health issue, would make a case towards a better understanding of the relevance of measuring it and including it in studies like the one you have made.
Finally, I would recommend to cite a couple more references of IJERPH or MDPI, since your research has impact in Global South countries, and the access to open data base is, in most cases, the only way to access this level of research, it would help scholars in the regions that interest you to achieve a more comprehensive knowledge of your contribution.
Author Response
We are grateful to the reviewer for taking their time to read the manuscript. We received invaluable comments and feedback from the reviewer. In the following lines, we try to highlight how we addressed all comments raised by the reviewer.
Note: our responses are written in blue.
Reviewer: Very interesting and relevant study, with high possibilities to be replicated in other parts of the world.
My only concern is that MEPI does not include any energy services, or considerations to energy deprivations, related to thermal comfort. I think that acknowledging the importance of thermal comfort as a public health issue, would make a case towards a better understanding of the relevance of measuring it and including it in studies like the one you have made.
Authors: we acknowledge thermal comfort as public health issue. However, this study is conducted in sub-Saharan Africa where public health concern is deprivation from clean energy sources. We also appreciate the difference in energy poverty measurement approaches/methods in colder and warmer parts of the world.
We have also made a comment on this in the Discussion. Energy services are very limited in rural Uganda.
Reviewer: Finally, I would recommend citing a couple more references of IJERPH or MDPI, since your research has impact in Global South countries, and the access to open data base is, in most cases, the only way to access this level of research, it would help scholars in the regions that interest you to achieve a more comprehensive knowledge of your contribution.
Authors: We added references as suggested.
Reviewer 3 Report
The authors examined an important problem with quantitative tools, which is the impact of microenvironmental risk factors and women empowermental factors on the health of children under five as measured by the incidence of three selected diseases. They constructed the multidimentional energy poverty index and the three-dimentional women empowerment index. The study also took into account the factors influencing the health of children, such as the source of drinking water, sanitation facility, hygiene facility.
The following were used for statistical analysis: descriptive statistics, bivariate analysis and logistic regression models.
The article deals with one of the most important problems of LMIC countries on the example of Uganda. The authors refer to the UN SDGs in an inconsistent, random manner. Therefore, one may ask on what basis, in the sixth point, the authors reach conclusions regarding changes in the scope of SDG 7, as well as other conclusions related to selected (why these?) SPGs.
It seems worth mentioning the results of similar studies carried out in other countries. There is only one reference to Bangladesh. Is it possible to show the changes of the studied problem that occurred over time in LMIC countries?
The variables are clearly described, but it would be worth presenting a more complete description of the research tools used, especially the regression models, in order to convey the course of the study in more detail.
The summary is very general and short. Shouldn't it be based on the results obtained? The conclusions regarding the SDGs appear to be not entirely justified.
Author Response
Reviewer 3
We are grateful to the reviewer for taking their time to read the manuscript. We received invaluable comments and feedback from the reviewer. In the following lines, we try to highlight how we addressed all comments raised by the reviewer.
Note: our responses are written in blue.
Reviewer: The authors examined an important problem with quantitative tools, which is the impact of microenvironmental risk factors and women empowerment factors on the health of children under five as measured by the incidence of three selected diseases. They constructed the multidimensional energy poverty index and the three-dimensional women empowerment index. The study also considered the factors influencing the health of children, such as the source of drinking water, sanitation facility, hygiene facility.
The following were used for statistical analysis: descriptive statistics, bivariate analysis and logistic regression models. The article deals with one of the most important problems of LMIC countries on the example of Uganda. The authors refer to the UN SDGs in an inconsistent, random manner.
Therefore, one may ask on what basis, in the sixth point, the authors reach conclusions regarding changes in the scope of SDG 7, as well as other conclusions related to selected (why these?) SDGs.
Authors: We omitted some of the text on SDGs and improved the wording in the remaining texts and
are more modest. Thank you.
Reviewer: It seems worth mentioning the results of similar studies carried out in other countries. There is only one reference to Bangladesh. Is it possible to show the changes of the studied problem that occurred over time in LMIC countries?
Authors: We performed searches. The Bangladesh references is the only one relevant and comprehensive enough for our paper. We have also added more references suggested by reviewer 4.
Reviewer: The variables are clearly described, but it would be worth presenting a more complete description of the research tools used, especially the regression models, in order to convey the course of the study in more detail.
Authors: The regression model we used in this study is fairly used in most disciplines and we presented in a simplified way, but concisely. If it is a relatively new model, it would be good to present it in detail.
Reviewer: The summary is very general and short. Shouldn't it be based on the results obtained? The conclusions regarding the SDGs appear to be not entirely justified.
Authors: We included a substantial number of relevant outcomes in the abstract. We omitted the reference to the SDGs in the conclusion as our outcomes don’t cover the SDGs entirely, of course but indicate in the text when our findings are relevant. We do see our work as guided by the SDG framework. We kept our reference to it in the introduction and discussion yet rephrased it.
Reviewer 4 Report
There are some major revisions that author(s) should made:
The authors of the research paper “Household microenvironment and under-fives health outcomes in Uganda: a comprehensive analysis including the multidi-mensional energy poverty and women empowerment” presented a topic very relevant to women empowerment.
First of all, I would like to point out that materials, method and empirical results and statistics have been properly realized and the research methodology is appropriate.
Probably conclusions should be extended in order to provide with a more detailed final idea.
My main concern is about the introduction and especially the references. As many researchers have studied this topic from different views and related another item, I suggest you to expand your international references on the subject with some ones below and not only with Uganda references in order to provide the article with a wide range of international references as below.
Although I know this is not the key point of the paper, I miss some references in the paper on this subject, also very relevant to women empowerment. Let me suggest some very recently references on this topic.: Santandreu, E.M.; López Pascual, J.; Cruz Rambaud, S. (2020) Determinants of Repayment among Male and Female Microcredit Clients in the USA. An Approach Based on Managers’ Perceptions. Sustainability, 12, 1701. doi: 10.3390/su12051701; and D’Espallier, B.; Guérin, I.; Mersland, R. Women and Repayment in Microfinance: A Global Analysis. World, Dev. 2011, 39, 758–772
The authors (page 12) say that “…Another important result for the analyses is the strong and consistent association be-tween women empowerment and child health outcomes considered in this study. The results revealed that women empowerment domains, particularly, freedom from domes-tic violence and social independence of women, were vital to reduce prevalence of ARI, diarrhea, and stunting among children.…” this is not very clear to me and it should be explained in a much more detailed way. Also, please, pay attention to typewriting mistakes such as “diarrhea” above.
Besides the discussion and conclusions should be rewritten because the conclusions are very weak developed. So I do suggest to the authors to rewrite it including some limitations and future researches.
I think that Extensive editing of English language and style required.
Therefore, my opinion is a major revision of the whole paper.
Author Response
Reviewer 4
We are grateful to the reviewer for taking their time to read the manuscript. We received invaluable comments and feedback from the reviewer. In the following lines, we try to highlight how we addressed all comments raised by the reviewer.
Note: our responses are written in blue.
Reviewer: There are some major revisions that author(s) should made: The authors of the research paper “Household microenvironment and under-fives health outcomes in Uganda: a comprehensive analysis including the multidimensional energy poverty and women empowerment” presented a topic very relevant to women empowerment.
First of all, I would like to point out that materials, method and empirical results and statistics have been properly realized and the research methodology is appropriate. Probably conclusions should be extended in order to provide with a more detailed final idea.
Authors: We added some components.
Reviewer: My main concern is about the introduction and especially the references. As many researchers have studied this topic from different views and related another item, I suggest you expand your international references on the subject with some ones below and not only with Uganda references in order to provide the article with a wide range of international references as below.
Although I know this is not the key point of the paper, I miss some references in the paper on this subject, also very relevant to women empowerment. Let me suggest some very recently references on this topic.:
Santandreu, E.M.; López Pascual, J.; Cruz Rambaud, S. (2020) Determinants of Repayment among
Male and Female Microcredit Clients in the USA. An Approach Based on Managers’ Perceptions. Sustainability, 12, 1701. doi: 10.3390/su12051701;
D’Espallier, B.; Guérin, I.; Mersland, R. Women and Repayment in Microfinance: A Global Analysis. World, Dev. 2011, 39, 758–772
Authors: thank you for the suggestion. We added more citations of women empowerment and health related papers.
Reviewer: The authors (page 12) say that “...Another important result for the analyses is the strong and consistent association between women empowerment and child health outcomes considered in this study. The results revealed that women empowerment domains, particularly, freedom from domes-tic violence and social independence of women, were vital to reduce prevalence of ARI, diarrhea, and stunting among children....” this is not very clear to me, and it should be explained in a much more detailed way. Also, please, pay attention to typewriting mistakes such as “diarrhoea” above.
Authors: Louise N. (native English speaker) Edited the whole paper. We used the UK spelling and grammar checking software, but kept the US and Canadian spelling in the references.
Reviewer: Besides the discussion and conclusions should be rewritten because the conclusions are very weak developed. So, I do suggest to the authors to rewrite it including some limitations and future research.
I think that Extensive editing of English language and style required.
Authors: We used the UK spell and grammar check software, but kept the US and Canadian spelling in the references. The conclusion and discussion are also revised. Therefore, my opinion is a major revision of the whole paper.
Authors: We agree. The new version improved the quality of the paper.
Round 2
Reviewer 4 Report
Some of my concerns have not been attended by the authors, especially the references.
So, again:
“As many researchers have studied this topic from different views and related another item, I suggest you expand your international references on the subject with some ones below and not only with Uganda references in order to provide the article with a wide range of international references as below. Although I know this is not the key point of the paper, I miss some references in the paper on this subject, also very relevant to women empowerment.
Let me suggest some very recently references on this topic.: Santandreu, E.M.; López Pascual, J.; Cruz Rambaud, S. (2020) Determinants of Repayment among Male and Female Microcredit Clients in the USA. An Approach Based on Managers’ Perceptions. Sustainability, 12, 1701. doi: 10.3390/su12051701; D’Espallier, B.; Guérin, I.; Mersland, R. Women and Repayment in Microfinance: A Global Analysis. World, Dev. 2011, 39, 758–772 Authors: thank you for the suggestion”.
Although the authors say that they have added more citations of women empowerment and health related papers, in fact none of previous references have been included.
Finally, and again the conclusions should be extended in order to provide with a more detailed final idea, probably it should be written again.
Author Response
Second round comments and responses- Reviewer 4
Some of my concerns have not been attended by the authors, especially the references.
Authors: We thank you for your interest in the topic and taking your time look into the paper once again. Apologies for the confusion on the concerns. I have gain made effort to address your concerns, in sensible way.
Reviewer: So, again: “As many researchers have studied this topic from different views and related another item, I suggest
you expand your international references on the subject with some ones below and not only with Uganda references in order to provide the article with a wide range of international references as below. Although I know this is not the key point of the paper, I miss some references in the paper on this subject, also very relevant to women empowerment.
Let me suggest some very recently references on this topic.: Santandreu, E.M.; López Pascual, J.; Cruz Rambaud, S. (2020) Determinants of Repayment among Male and Female Microcredit Clients in the USA.
An Approach Based on Managers’ Perceptions. Sustainability, 12, 1701. doi: 10.3390/su12051701; D’Espallier, B.; Guérin, I.; Mersland, R. Women and Repayment in Microfinance: A Global Analysis. World, Dev. 2011, 39, 758–772 Authors: thank you for the suggestion”.
Although the authors say that they have added more citations of women empowerment and health related papers, in fact none of previous references have been included.
Authors: We agree the topic is widely researched from different perspectives. We tended to focus on women empowerment and its relations with health outcomes of young children. We have once again made effort to search literature on women empowerment from microfinance perspective. We have included three references that are relevant to our topic. In citing those more papers, we considered them for their theoretical relevance (if the paper gives theoretical foundation to our work), for methodological relevance, or if those papers are related to our topic.
We also appreciated your suggestion of the recent paper to be cited. This paper is more recent, but it focusses on the USA, it refers to the global literature and also it is referencing to the first referenced paper by D’Espallier et al, and it has no global implications. Nor it does include the women empowerment index.
We tend not include this paper in the references as it merely uses loosely the global literature to state “ In the USA, there are not—as in other countries—strong incentives, motivations, or external pressures, other than those that men also have, which influence women to pay their microloans better than men. Then, domestic and international MFIs attracted to enter the USA’s market should review their microcredit policies in relation to women.”
We hope the reviewer recognizes the suggested papers are more on credit worthiness of women compared with men. The papers neither deal with women empowerment nor its impact on health, wellbeing (or any other outcome).
Reviewer: Finally, and again the conclusions should be extended in order to provide with a more detailed final idea, probably it should be written again.
Authors: We expected the reviewer have recognized significant changes we made in the first-round comments. In the round 1, we did already restructure the Discussion rather rigorously, referring to the findings and making it more focussed, leading logically to our general conclusion. We have again worked on the conclusion section. The conclusion is restructured. Few details are also included. However, we don’t believe our multi-determinant approach and the topic can lead to a (single) final idea. It now summarizes
both our findings, on energy and women’s empowerment, better.